# A [3+3] Aldol-S_N_Ar-Dehydration Approach to 2-Naphthol and 7-Hydroxyquinoline Derivatives

**DOI:** 10.3390/molecules29143406

**Published:** 2024-07-20

**Authors:** Kwabena Fobi, Ebenezer Ametsetor, Richard A. Bunce

**Affiliations:** Department of Chemistry, Oklahoma State University, Stillwater, OK 74078-3071, USA; kfobi@okstate.edu (K.F.); eametse@okstate.edu (E.A.)

**Keywords:** hindered 2-naphthols, [3+3] annulation, sequential reactions, aldol-S_N_Ar-dehydration, ring formation

## Abstract

A one-pot [3+3] aldol-S_N_Ar-dehydration annulation sequence was utilized to fuse hindered phenols onto aromatic substrates. The transformation joins doubly activated 1,3-disubstituted acetone derivatives (dinucleophiles) with C5-activated 2-fluorobenzaldehyde S_N_Ar acceptors (dielectrophiles) in the presence of K_2_CO_3_ in DMF at 65–70 °C to form polysubstituted 2-naphthols and 7-hydroxyquinolines. The reaction is regioselective in adding the most stable anionic center to the aldehyde followed by S_N_Ar closure of the less stabilized anion to the electron-deficient aromatic ring. Twenty-seven examples are reported, and a probable mechanism is presented. In two cases where S_N_Ar activation on the acceptor ring was lower (a C5 trifluoromethyl group on the aromatic ring or a 2-fluoropyridine), diethyl 1,3-acetonedicarboxylate initiated an interesting Grob-type fragmentation to give cinnamate esters as the products.

## 1. Introduction

An earlier report from the same laboratory used in this study outlined a [3+3] annulation to prepare 4*H*-1-benzopyrans [1]. Over the past decade, there have been many additional examples that utilized this strategy to generate a variety of complex molecules. We summarized a number of these reports in a recently published study that used a [3+3] approach to access quinolin-2(1*H*)-ones and 1,8-naphthyridine-2(1*H*)-ones [2]. The current work extends this method to the preparation of polysubstituted 2-naphthols and 7-hydroxyquinolines.

There are many commercially available 2-naphthol derivatives available at a relatively low cost, and synthetic transformations on the hydroxylated ring are quite facile. In addition to activating the OH-bearing ring through electron donation, the tautomeric nature of the OH group at C2 imparts significant nucleophilicity to C1. Thus, alkylations [3,4,5] and condensations [6,7,8,9,10] at C1 can be carried out under relatively mild conditions. Additionally, halogenation [11,12] and amination [13] procedures offer further strategies for functionalization at C1.

It is well established that 2-naphthol is an important scaffold in organic chemistry. It is a precursor to the BINOL system, which encompasses an important family of chiral ligands and catalysts [14] as well as some structures of biological importance [15,16]. Additionally, 2-naphthol is important in the synthesis of a wide variety of heterocyclic systems via multicomponent reactions [17,18].

Most modern syntheses of 2-naphthols involve the use of metal-promoted electrophilic [19,20,21], coupling [22], cycloaddition [23], or radical [24] approaches to effectively insert an ethene fragment between the terminus of a two-carbon side chain and the ortho carbon of an aryl ring. The current reaction permits the annulation of a substituted phenol to an appropriately substituted aromatic substrate. The process involves the stepwise addition of the two anions derived from a 1,3-disubstituted acetone (dinucleophile) to a C5-substituted 2-fluorobenzaldehyde derivative (dielectrophile) in a one-pot aldol-S_N_Ar-dehydration sequence (see Figure 1). This provides a straightforward synthesis of 2-naphthols without metal catalysts or other additives. This minimizes reaction optimization and problems associated with post-reaction metal contamination. One additional point of interest is the potential regioselectivity of the reaction when unsymmetrically substituted 1,3-disubstituted acetones are employed.

Recently, several 2-naphthol-based drug candidates have been reported in the literature. These derivatives expressed significant activity against cancer as well as Gram-positive and Gram-negative bacteria. While many active compounds have been disclosed, several of the most potent are depicted in Figure 1. Compound **1** demonstrated excellent activity (IC_50_ ≤ 1.2 μM) against Hep G2 (liver), A549 (lung), MDA-231 (breast), and HeLa (cervical) cancer cell lines [25]. Derivative **2** showed impressive GI_50_ values at concentrations of 1.46–2.90 μM against different strains of lung and breast cancers [26]. Compound **3** was found to inhibit 17β-hydroxysteroid dehydrogenases (17β-HSD1–89%) and 17β-HSD2 (61%) at a 1 μM concentration, which could favorably impact the treatment of estrogen-dependent diseases such as breast cancer and endometriosis [27]. Finally, structure **4** showed an IC_50_ value of 0.5 μg/mL in screens with *S. aureus* (Gram-positive) and a value of 2 μg/mL with *E. coli* (Gram-negative) bacteria [10].

## 2. Results and Discussion

The results of our study are summarized in Figure 2 and Table 1, Table 2, Table 3 and Table 4. The Appendix A give the ^1^H NMR and ^13^C NMR spectra for all new compounds. The reaction proceeded in the highest yield when 1 equiv. of the 2-fluorobenzaldehyde derivative (dielectrophile) was reacted with 2 equiv. of the 1,3-disubstituted acetone (dinucleophile) in the presence of 2 equiv. of anhydrous K_2_CO_3_ in dry DMF at 65–70 °C. Other combinations (1:1, 1.5:1, 2:1, and 1:1.5) of dielectrophile:dinucleophile gave lower yields. The reactants were mixed at room temperature (23 °C) and heated to 65–70 °C prior to the addition of the base. Further heating at this temperature for 6–12 h completed the reaction. Mild aqueous acid work-up and silica gel column chromatography (10–20% EtOAc in hexane) then delivered the purified products. C5 activation of the 2-fluorobenzaldehyde included resonance withdrawing (NO_2_, CN) and inductively withdrawing (CF_3_) groups; 2-fluoronicotinaldehyde was also explored as a dielectrophile. The 1,3-disubstituted acetones were substituted by ester, phenyl ketone, or phenylsulfonyl groups adjacent to the most acidic methylene with ester, phenyl ketone, or phenyl adjacent to the equivalent or less acidic site. Alkyl ketones failed in the reaction presumably due to their competitive enolization under the strong basic conditions. Successful examples proceeded in a high yield in a single laboratory operation. The reaction temperature and run time were based on our previous report [2]. The reactions were monitored by TLC and judged to be complete when all of the 2-fluorobenzaldehyde dielectrophile was consumed.

The 2-fluoroarylaldehyde derivatives **5a**–**d** were commercially available, as were dimethyl and diethyl 1,3-acetonedicarboxylates **6** and **7**, respectively. Other 1,3-disubstituted acetones were obtained by the use or adaptation of synthetic processes in the literature. The various methyl 4-aryl-3-oxobutanoates (**8**–**14**) were prepared by the general procedure reported for the synthesis of 4-phenyl-3-oxobutanoate by Yonemitsu and co-workers [28]. The method of Hauser et al. was used to access 1,3-dibenzoylacetone (**15**) [29]. The preparation of 1-phenyl-3-(phenylsulfonyl)propan-2-one (**17**) was accomplished by adapting the strategy developed by Nájera’s group for the synthesis of 1-phenyl-3-(*p*-tosyl)propan-2-one [30,31]. Finally, the synthesis of bis(benzenesulfonyl)propan-2-one (**18**) was accomplished using the procedure of Poli and co-workers [32].

An interesting feature of the reaction was the regioselectivity observed when unsymmetrical 1,3-disubstituted acetone derivatives were employed. For the substrates studied, the most acidic methylene was observed to attack the aldehyde, leaving the less acidic site to react with the aromatic ring in the final S_N_Ar ring closure [33]. The process was promoted by K_2_CO_3_; no other catalysts or additives were required. Thus, the reactions predictably afforded a single product, making the yields higher and the final purifications less tedious.

A probable mechanism for the current reaction is given in Figure 3 for the reaction of 2-fluoro-5-nitrobenzaldehyde (**1a**) with methyl 4-phenyl-3-oxobutanoate (**8**). The addition of the more stable β-ketoester anion to the aldehyde would give aldol product **A**. The subsequent deprotonation and reaction of the less acidic α-carbon at the fluorine-bearing site on the S_N_Ar acceptor would close the ring to afford **B**. As previous work has demonstrated [2], this process requires a base to form the anion and an elevated temperature to promote the S_N_Ar ring closure. The elimination of water from the initial ring-closed product **B** would then give enone **C** and tautomerization to the aromatic 2-naphthol **20**. The late-stage loss of water would assure a favorable geometry for the ring closure of the anion derived from **A**.

In general, nitro-activated S_N_Ar acceptor **5a** gave the best results, followed by cyano-substituted substrate **5b**. Dielectrophiles 5-trifluoromethyl-2-fluorobenzaldehyde (**5c**) and 2-fluoronicotinaldehyde (**5d**) were less reactive but also afforded annulation products with several of the dinucleophiles. Only one dinucleophile, 1,3-bis(phenylsulfonyl)propan-2-one (**17**), failed to give a [3+3] product in the nitro series (Table 1, entry 12). For this substrate, exposure to K_2_CO_3_ at 65–70 °C resulted in the decomposition of the bis(sulfone). Treatment with a milder tertiary amine base (Et_3_N) at the same temperature also resulted in the degradation of **17** but led to a respectable yield (82%) of 5-nitro-2-(phenylsulfonyl)benzaldehyde (**29**). The confirmation of the identity of **29** was accomplished by comparison with the same compound prepared from 2-fluoro-5-nitrobenzaldehyde and 2 equiv. of sodium benzenesulfinate in DMF containing 2 equiv. of Et_3_N at 65–70 °C for 30 min.

Finally, two examples involving reactions of the less-activated S_N_Ar acceptors **5c** and **5d** with diethyl 1,3-acetonedicarboxylate (**7**) underwent elimination reactions to afford cinnamate ester products rather than the expected S_N_Ar cyclization to produce 2-naphthol **37** and 7-hydroxyquinoline **40**, respectively (entry 1 in Table 3 and Table 4). In these transformations, one can envision a base-initiated Grob-type fragmentation [34] (Figure 4) with the loss of a malonate fragment in addition to hydroxide. The antiperiplanar alignment of these groups should facilitate this elimination to the stable conjugated ester product.

## 3. Materials and Methods

### 3.1. General Methods

Unless otherwise indicated, all reactions were performed under dry N_2_ in dry glassware. All commercial reagents and solvents were used as received (Combi Blocks, San Diego, CA, USA and Fisher Scientific, Pittsburgh, PA, USA). All wash solutions employed in work-up procedures were aqueous. Reactions were followed by thin layer chromatography on Analtech No 21521 silica gel GF plates (Newark, DE, USA). Preparative separations were accomplished by column chromatography on Davisil^®^, grade 62, 60–200-mesh silica gel containing 0.5% of UV-05 phosphor (both from Sorbent Technologies, Norcross, GA, USA) slurry packed into quartz columns. Band elution for all chromatographic separations was monitored using a hand-held ultraviolet lamp (Fisher Scientific, Pittsburgh, PA, USA). Melting points (uncorrected) were obtained using a MEL-TEMP apparatus (Cambridge, MA, USA). FT-IR spectra were run as thin films on sodium chloride disks or in ATR mode using a Nicolet iS50 spectrophotometer (Madison WI, USA). ^1^H- and ^13^C-NMR spectra were obtained using a Bruker Avance 400 system (Billerica, MA, USA) at 400 MHz and 101 MHz, respectively, in CDCl_3_ or DMSO-*d_6_* containing 0.05% tetramethylsilane as the internal standard (Cambridge Isotope Laboratories, Andover, MA, USA). Chemical shifts are given in ppm downfield from the internal standard, and coupling constants (*J*) are reported in Hz. Low-resolution mass spectra were obtained using a Hewlett-Packard Model 1800A GCD GC-MS system (Palo Alto, CA, USA). Elemental analyses (±0.4%) on all new compounds were carried out by Atlantic Microlabs (Norcross, GA, USA).

β-Ketoesters **8**–**14** were prepared from 5 g each of the substituted phenylacetyl chloride derivatives according to the procedure used by Yonemitsu and co-workers [29]. 1,3-Dibenzoylacetone (**15**) was prepared on a 25 mmol scale using the method by Hauser et al. [30]. The remaining 1,3-disubstituted acetones required multistep procedures. Several of these dinucleophiles are known, but the spectra were not always provided in the past. Some contain varying amounts of the corresponding enols.

### 3.2. Methyl 3-Oxo-4-phenylbutanoate (***8***)

Yield: 5.09 g (82%) as a colorless oil, b.p. 92–96 °C at 0.05 mm Hg (reference [28], b.p. 126–128 °C at 0.6 mm Hg); IR: 1745, 1721 cm^−1^; ^1^H NMR (400 MHz, CDCl_3_): δ 7.37–7.25 (complex, 3H), 7.21 (dm, *J* = 7.4 Hz, 2H), 3.82 (s, 2H), 3.71 (s, 3H), 3.47 (s, 2H); ^13^C NMR (101 MHz, CDCl_3_): δ 200.4, 167.6, 133.2, 129.6, 128.9, 127.4, 52.4, 50.1, 48.0; MS (*m*/*z*): 192 (C_11_H_12_O_3_, M^+^).

### 3.3. Methyl 4-(2-Fluorophenyl)-3-oxobutanoate (***9***)

Yield: 4.14 g (68%) as a colorless oil, b.p. 95–98 °C (0.08 mm Hg); IR: 1746, 1722, 1235 cm^−1^; ^1^H NMR (400 MHz, CDCl_3_): δ 7.27 (m, 1H), 7.19 (td, *J* = 7.6, 1.9 Hz, 1H), 7.13–7.03 (complex, 2H), 3.87 (s, 2H), 3.72 (s, 3H), 3.53 (s, 2H); ^13^C NMR (101 MHz, CDCl_3_): δ 199.2 167.5, 161.0 (d, *J* = 246.1 Hz), 131.8 (d, *J* = 4.1 Hz), 129.4 (d, *J* = 6.5 Hz), 124.4 (d, *J* = 3.6 Hz), 120.7 (d, *J* = 16.1 Hz), 115.5 (d, *J* = 21.7 Hz), 52.4, 48.2, 43.1 (d, *J* = 2.5 Hz); MS (*m*/*z*): 210 (C_11_H_11_FO_3_, M^+^).

### 3.4. Methyl 4-(4-Chlorophenyl)-3-oxobutanoate (***10***)

Yield: 4.19 g (70%) as a colorless oil, b.p. 117–121 °C (0.01 mm Hg); IR: 1748, 1725 cm^−1^; ^1^H NMR (400 MHz, CDCl_3_): δ 7.31 (d, *J* = 8.5 Hz, 2H), 7.14 (d, *J* = 8.5 Hz, 2H), 3.81 (s, 2H), 3.73 (s, 3H), 3.47 (s, 2H); ^13^C NMR (101 MHz, CDCl_3_): δ 199.7, 167.4, 133.4, 131.5, 130.9, 129.0, 52.4, 49.1, 48.2; MS (*m*/*z*): 226, 228 (C_11_H_11_ClO_3_, ca. 3:1 M^+^).

### 3.5. Methyl 4-(4-Methylphenyl)-3-oxobutanoate (***11***)

Yield: 4.52 (74%) as a colorless oil, b.p. 104–107 °C (0.01 mm Hg); IR: 1745, 1719 cm^−1^; ^1^H NMR (400 MHz, CDCl_3_): δ 7.15 (d, *J* = 8.1 Hz, 2H), 7.09 (s, *J* = 8.1 Hz, 2H), 3.77 (s, 2H), 3.71 (s, 3H), 3.44 (s, 2H), 2.33 (s, 3H); ^13^C NMR (101 MHz, CDCl_3_): δ 200.7, 167.6, 137.1, 130.1, 129.6, 129.4, 52.4, 49.7, 47.9, 21.1; MS (*m*/*z*): 206 (C_12_H_14_O_3_, M^+^).

### 3.6. Methyl 4-(4-Methoxyphenyl)-3-oxobutanoate (***12***)

Yield: 4.81 g (80%) as a colorless oil, b.p. 121–25 °C (0.01 mm Hg); IR: 2839, 1747, 1724 cm^−1^; ^1^H NMR (400 MHz, CDCl_3_): δ 7.12 (d, *J* = 8.6 Hz, 2H), 6.88 (d, *J* = 8.6 Hz, 2H), 3.80 (s, 3H), 3.76 (s, 2H), 3.71 (s, 3H), 3.45 (s, 2H); ^13^C NMR (101 MHz, CDCl_3_): δ 200.8, 167.3, 158.9, 130.6, 125.1, 114.3, 55.3, 52.4, 49.2, 47.8; MS (*m*/*z*): 222 (C_12_H_14_O_4_, M^+^).

### 3.7. Methyl 3-Oxo-4-(3-(trifluoromethyl)phenyl)butanoate (***13***)

Yield: 3.62 g (62%) as a colorless oil, b.p. 121–125 °C (0.01 mm Hg); IR: 1749, 1725 cm^−1^; ^1^H NMR (400 MHz, CDCl_3_): δ 7.55 (m, 1H), 7.44 (m, 2H), 7.40 (m, 1H), 3.92 (s, 2H), 3.71 (s, 3H), 3.52 (s, 2H); ^13^C NMR (101 MHz, CDCl_3_): δ 199.6, 167.5, 134.1, 133.2, 129.2, 126.4 (q, *J* = 3.9 Hz), 124.1 (q, *J* = 4.0 Hz), 124.0 (q, *J* = 272.3 Hz), 52.4, 49.1, 48.3; MS (*m*/*z*): 260 (C_12_H_11_F_3_O_3_, M^+^).

### 3.8. Methyl 4-(2,5-Dimethylphenyl)-3-oxobutanoate (***14***)

Yield: 5.12 g (85%) as a colorless oil that solidified to a waxy solid, b.p. 108–112 °C (0.001 mm Hg); IR: 1747, 1720 cm^−1^; ^1^H NMR (400 MHz, CDCl_3_): δ 7.07 (d, *J* = 7.7 Hz, 1H), 7.01 (dd, *J* = 7.7, 1.9 Hz, 1H), 6.94 (d, *J* = 1.9 Hz, 1H), 3.79 (s, 2H), 3.71 (s, 3H), 3.44 (s, 2H), 2.30 (s, 3H), 2.19 (s, 3H); ^13^C NMR (101 MHz, CDCl_3_): δ 200.5, 167.6, 135.9, 133.9, 131.8, 131.2, 130.5, 128.4, 52.3, 48.4, 47.9, 20.9, 19.1; MS (*m*/*z*): 220 (C_13_H_16_O_3_, M^+^).

### 3.9. 1,3 Dibenzoylpropan-2-One (***15***)

Yield: 5.32 g (80%) as a yellow solid, m.p. 105–108 °C (lit [29] m.p. 105–109 °C); IR: 3194–2990, 1682, 1600, 1552, 1496 cm^−1^; ^1^H NMR (400 MHz, CDCl3): δ 14.76 (s, 2H), 7.86 (dm, *J* = 6.9 Hz, 4H), 7.53–7.42 (complex, 6H), 6.02 (s, 2H); ^13^C NMR (101 MHz, CDCl_3_): δ 194.1, 173.7, 133.7, 131.7, 128.6, 126.4, 96.8; MS (*m*/*z*): 266 (C_17_H_14_O_3_, M^+^).

### 3.10. 1-Phenyl-3-(Phenylsulfonyl)propan-2-one (***16***)

Yield: 0.88 g (32%, 3 steps) as a white solid, m.p. 82–83 °C; IR: 1716, 1314, 1136 cm^−1^; ^1^H NMR (400 MHz, CDCl_3_): δ 7.88 (d, *J* = 7.9 Hz, 2H), 7.67 (m, 1H), 7.57 (t, *J* = 7.9 Hz, 2H), 7.36–7.26 (complex, 3H), 7.19 (d, *J* = 7.9 Hz, 2H), 4.16 (s, 2H), 3.98 (s, 2H); ^13^C NMR (101 MHz, CDCl_3_): δ 195.7, 134.4, 129.7, 129.4, 129.1, 128.7, 128.3, 127.9, 127.7, 65.6, 50.9; MS (*m*/*z*): 274 (C_15_H_14_O_3_S, M^+^).

### 3.11. 1,3 Bis(Phenylsulfonyl)propan-2-one (***17***)

Yield: 0.56 g (10%) as an off-white solid, m.p. 156–158 °C (lit [32] m.p. 157–159 °C); IR: 1732, 1341, 1150 cm^−1^; ^1^H NMR (400 MHz, CDCl_3_): δ 7.85 (dm, *J* = 7.5 Hz, 4H), 7.71 (tt, *J* = 7.5, 1.8 Hz, 2H), 7.58 (tt, *J* = 7.5, 1.8 Hz, 4H), 4.51 (s, 4H); ^13^C NMR (101 MHz, CDCl_3_): δ 186.4, 138.2, 134.7, 129.6, 128.3, 67.6; MS (*m*/*z*): 338 (C_15_H_14_O_5_S_2_, M^+^).

### 3.12. Representative Procedure for the [3+3] Preparation of Hindered 2-Naphthols

The 1,3-disubstituted acetone (2 equiv.) was added to a solution of the C5-substituted 2-fluorobenzaldehyde derivative (1.0–1.2 mmol, 1 equiv.) in DMF (10 mL) under N_2_. The mixture was stirred for 5 min at 23 °C and then for 15 min at 65–70 °C prior to the addition of anhydrous K_2_CO_3_ (2 equiv.). Heating was continued until TLC (20% EtOAc in hexane) indicated complete consumption of the starting materials (6–12 h). The mixture was poured into water (30 mL) and extracted with EtOAc (3 × 30 mL). The combined organic extracts were washed with 1 M HCl (2 × 30 mL), saturated NaHCO_3_ (1 × 30 mL), and saturated NaCl (1 × 30 mL), and then dried (Na_2_SO_4_) and filtered. The crude product was concentrated under a vacuum, and the resulting oil/solid was purified by column chromatography eluted with 10–20% EtOAc in hexane to give the pure products. The following compounds were prepared:

#### 3.12.1. Dimethyl 2-Hydroxy-6-nitronaphthalene-1,3-dicarboxylate (**18**)

Yield: 0.32 g (89%) as a white solid, m.p. 187–189 °C; IR: 3240–2980, 1726, 1683, 1533, 1342 cm^−1^; ^1^H NMR (400 MHz, CDCl_3_): δ 11.59 (s, 1H), 8.80 (d, *J* = 2.4 Hz, 1H), 8.74 (s, 1H), 8.33 (dd, *J* = 9.4, 2.4 Hz, 1H), 8.03 (d, *J* = 9.4 Hz, 1H), 4.10 (s, 3H), 4.09 (s, 3H); ^13^C NMR (101 MHz, CDCl_3_): δ 168.8, 167.1, 158.1, 144.2, 137.0, 136.9, 126.4, 125.7, 124.9, 123.5, 116.7, 115.8, 53.4, 53.0; MS *m*/*z*: 305 (M^+^); Anal. Calcd for C_14_H_11_NO_7_: C, 55.09; H, 3.63; N, 4.59; found: C, 55.12; H, 3.64; N 4.51.

#### 3.12.2. Diethyl 2-Hydroxy-6-nitronaphthalene-1,3-dicarboxylate (**19**)

Yield: 0.35 g (88%) as a white solid, m.p. 130–131 °C; IR: 3435, 1722, 1504, 1334 cm^−1^; ^1^H NMR (400 MHz, CDCl_3_): δ 11.62 (s, 1H), 8.82 (d, *J* = 2.4 Hz, 1H), 8.74 (s, 1H), 8.33 (dd, *J* = 9.4, 2.4 Hz, 1H), 8.02 (d, *J* = 9.4 Hz, 1H), 4.55 (overlapping q, *J* = 7.1 Hz, 4H), 1.51 (overlapping t, *J* = 7.1 Hz, 6H); ^13^C NMR (101 MHz, CDCl_3_): δ 168.5, 166.6, 158.0, 144.1, 136.8, 136.6, 126.4, 125.6, 124.9, 123.3, 116.9, 116.3, 62.8, 62.2, 14.3, 14.2; MS: (*m*/*z*): 333 (M^+^); Anal. Calcd for C_16_H_15_NO_7_: C, 57.66; H, 4.54; N, 4.20; found: C, 57.64; H, 4.52; N, 4.13.

#### 3.12.3. Methyl 3-Hydroxy-7-nitro-4-phenyl-2-naphthoate (**20**)

Yield: 0.32 g (89%) as a yellow solid, m.p. 201–203 °C; IR: 3300–2960, 1683, 1621, 1530, 1333 cm^−1^; ^1^H NMR (400 MHz, CDCl_3_): δ 11.10 (s, 1H), 8.83 (d, *J* = 2.4 Hz, 1H), 8.75 (s, 1H), 8.13 (dd, *J* = 9.4, 2.4 Hz, 1H), 7.60 (d, *J* = 9.4 Hz, 1H), 7.56 (t, *J* = 7.3 Hz, 2H), 7.49 (t, *J* = 7.3 Hz, 1H), 7.38 (d, *J* = 7.2 Hz, 2H), 4.09 (s, 3H); ^13^C NMR (101 MHz, CDCl_3_): δ 169.9, 156.4, 143.9, 139.1, 134.18, 134.16, 130.7, 128.7, 128.1, 126.7, 126.5, 125.1, 125.0, 122.0, 115.8, 53.2; MS (*m*/*z*): 323 (M^+^); Anal. Calcd for C_18_H_13_NO_5_: C, 66.87; H, 4.05; N, 4.33; found: C, 66.79; H, 4.01; N, 4.26.

#### 3.12.4. Methyl 4-(2-Fluorophenyl)-3-hydroxy-7-nitro-2-naphthoate (**21**)

Yield: 0.37 g (93%) as a light pink solid, m.p. 180–181 °C; IR: 3433, 1730, 1520, 1344, 1219 cm^−1^; ^1^H NMR (400 MHz, CDCl_3_): δ 13.09 (s, 1H), 8.90 (d, *J* = 9.6 Hz, 1H), 8.70 (d, *J* = 2.5 Hz, 1H), 8.33 (dd, *J* = 9.6, 2.5 Hz, 1H), 8.08 (s, 1H), 7.48–7.42 (complex, 2H), 7.30–7.20 (complex, 2H), 4.18 (s, 3H); ^13^C NMR (101 MHz, CDCl_3_): δ 172.4, 165.1, 160.1 (d, *J* = 248.5 Hz), 143.6, 138.8 (d, *J* = 1.5 Hz), 135.1, 131.6 (d, *J* = 3.1 Hz), 130.5 (d, *J* = 8.3 Hz), 129.1, 126.9, 126.7, 125.3, 124.1 (d, *J* = 3.7 Hz), 123.7 (d, *J* = 15.6 Hz), 121.8, 115.8 (d, *J* = 22.1 Hz), 105.4, 53.2; MS (*m*/*z*): 341 (M^+^); Anal. Calcd for C_18_H_12_FNO_5_: C, 63.35; H, 3.54; N, 4.10; found: C, 63.41; H, 3.55; N, 4.04.

#### 3.12.5. Methyl 4-(4-Chlorophenyl)-3-hydroxy-7-nitro-3-naphthoate (**22**)

Yield: 0.40 g (94%) as a yellow solid, m.p. 166–167 °C; IR: 3198–2735, 1691, 1504, 1336 cm^−1^; ^1^H NMR (400 MHz, DMSO-*d_6_*): δ 10.90 (s, 1H), 9.21 (d, *J* = 2.5 Hz, 1H), 9.05 (s, 1H), 8.21 (dd, *J* = 9.5, 2.5 Hz, 1H), 7.63 (d, *J* = 8.4 Hz, 2H), 7.50 (d, *J* = 9.5 Hz, 1H), 7.40 (d, *J* = 8.4 Hz, 2H), 4.04 (s, 3H); ^13^C NMR (101 MHz, CDCl_3_): δ 169.8, 156.4 144.0, 138.9, 134.5, 132.6, 132.1, 130.9, 129.0, 128.6, 126.6, 126.3, 125.0, 122.3, 115.9, 53.3; MS (*m*/*z*): 357, 359 (*ca*. 3:1, M^+^); Anal. Calcd for C_18_H_12_ClNO_5_: C, 60.43; H, 3.38; N, 3.92; found: C, 60.35; H, 3.36; N, 3.89.

#### 3.12.6. Methyl 3-Hydroxy-4-(4-methylphenyl)-7-nitro-3-naphthoate (**23**)

Yield: 0.36 g (91%) as an orange solid, m.p. 170–171 °C; IR: 3485, 1728, 1529, 1344 cm^−1^; ^1^H NMR (400 MHz, CDCl_3_): δ 8.16 (dd, *J* = 8.9 Hz, 1H), 8.00 (d, *J* = 2.7 Hz, 1H), 7.61 (d, *J* = 8.2 Hz, 2H), 7.21 (s, 1H), 7.18 (d, *J* = 8.2 Hz, 2H), 7.06 (d, *J* = 8.9 Hz, 1H), 6.66 (s, 1H), 3.91 (s, 3H), 2.37 (s, 3H); ^13^C NMR (101 MHz, CDCl_3_): δ 164.0, 158.4, 143.3, 143.1, 137.3, 131.6, 129.5, 129.3, 129.2, 127.6, 126.1, 123.4, 120.4, 115.6, 110.0, 52.7, 21.4; MS (*m*/*z*): 337 (M^+^); Anal. Calcd for C_19_H_15_NO_5_: C, 67.65; H, 4.48; N, 4.15; found: C, 67.57; H, 4.47; N, 4.11.

#### 3.12.7. Methyl 3-Hydroxy-4-(4-methoxyphenyl)-7-nitro-3-naphthoate (**24**)

Yield: 0.40 g (96%) as a light yellow solid, m.p. 198–199 °C; IR: 3404, 2844, 1729, 1518, 1337 cm^−1^; ^1^H NMR (400 MHz, CDCl_3_): δ 13.2 (s, 1H), 8.86 (d, *J* = 9.6 Hz, 1H), 8.70 (d, *J* = 2.5 Hz, 1H), 8.29 (dd, *J* = 9.6, 2.5 Hz, 1H), 8.05 (s, 1H), 7.59 (d, *J* = 8.8 Hz, 2H), 7.04 (d, *J* = 8.8 Hz, 2H), 4.18 (s, 3H), 3.89 (s, 3H); ^13^C NMR (101 MHz, CDCl_3_): δ 172.7, 165.3, 159.8, 143.6, 137.3, 134.5, 134.0, 130.7, 128.2, 127.2, 126.6, 125.1, 121.3, 113.9, 105.4, 55.4, 53.1; MS (*m*/*z*): 353 (M^+^); Anal. Calcd for C_19_H_15_NO_6_: C, 64.59; H, 4.28; N, 3.96; found: C, 64.52; H, 4.29; N, 3.93.

#### 3.12.8. Methyl 3-Hydroxy-7-nitro-4-(3-(trifluoromethyl)phenyl)-2-naphthoate (**25**)

Yield: 0.35 g (85%) as a white solid, m.p. 182–183 °C; IR: 3423, 1728, 1534, 1339 cm^−1^; ^1^H NMR (400 MHz, CDCl_3_): δ 13.23 (s, 1H), 8.90 (d, *J* = 9.6 Hz, 1H), 8.74 (d, *J* = 2.5 Hz, 1H), 8.34 (dd, *J* = 9.6, 2.5 Hz, 1H), 8.11 (s, 1H), 7.91 (s, 1H), 7.83 (d, *J* = 7.8 Hz, 1H), 7.71 (d, *J* = 7.8 Hz, 1H), 7.63 (t, *J* = 7.8 Hz, 1H), 4.20 (s, 3H); ^13^C NMR (101 MHz, CDCl_3_): δ 172.5, 164.7, 143.8, 137.9, 136.7, 134.9, 132.9, 132.8, 131.9 (q, *J* = 32.5 Hz), 128.9, 127.1, 126.8, 126.4 (q, *J* = 4.0 Hz), 125.3, 125.1 (q, *J* = 3.8 Hz), 124.1 (q, *J* = 272.5 Hz), 122.0, 105.7, 53.3; MS (*m*/*z*): 391 (M^+^); Anal. Calcd for C_19_H_12_F_3_NO_5_: C, 58.32; H, 3.09; N, 3.58; found: C, 58.26; H, 3.05; N, 3.61.

#### 3.12.9. Methyl 4-(2,5-Dimethylphenyl)-3-hydroxy-7-nitro-2-naphthoate (**26**)

Yield: 0.40 g (96%) as an orange solid, m.p. 180–181 °C; IR: 3439, 1726, 1528, 1345 cm^−1^; ^1^H NMR (400 MHz, CDCl_3_): δ 8.15 (dd, *J* = 9.0, 2.6 Hz, 1H), 8.02 (d, *J* = 2.6 Hz, 1H), 7.68 (s, 1H), 7.35 (s, 1H), 7.11 (d, *J* = 7.7 Hz, 1H), 7.00 (d, *J* = 7.7 Hz, 1H), 6.94 (d, *J* = 9.0 Hz, 1H), 6.85 (s, 1H), 3.92 (s, 3H), 2.37 (s, 3H), 2.33 (s, 3H); ^13^C NMR (101 MHz, CDCl_3_): δ 164.0, 158.4, 143.3, 143.0, 134.9, 133.8, 132.8, 130.2, 130.1, 129.3, 128.1, 127.7, 125.8, 123.4, 120.4, 115.6, 108.0, 52.7, 21.3, 20.0; MS (*m*/*z*): 351 (M^+^); Anal. Calcd for C_20_H_17_NO_5_: C, 68.37; H, 4.88; N, 3.99; found: C, 68.39; H, 4.88; N, 3.91.

#### 3.12.10. 1,3 Dibenzoyl-2-Hydroxy-6-nitronaphthalene (**27**)

Yield: 0.41 g (87%) as a light yellow solid, m.p. 140–141 °C; IR: 3280–2920, 1667, 1633, 1526, 1339 cm^−1^; ^1^H NMR (400 MHz, CDCl_3_): δ 11.71 (s, 1H), 8.81 (d, *J* = 2.3 Hz, 1H), 8.52 (s, 1H), 8.25 (dt, *J* = 9.4, 2.3 Hz, 1H), 7.93 (d, *J* = 8.5 Hz, 2H), 7.82 (d, *J* = 8.5 Hz, 2H), 7.74 (t, *J* = 7.4 Hz, 1H), 7.71–7.59 (complex, 4H), 7.50 (t, *J* = 7.6 Hz, 2H); ^13^C NMR (101 MHz, CDCl_3_): δ 200.9, 195.5, 157.6, 144.3, 139.6, 137.4, 137.1, 136.8, 134.3, 133.4, 129.7, 129.0, 128.9, 126.8, 125.5, 124.9, 123.7, 123.2, 122.1 (one carbon unresolved); MS (*m*/*z*): 397 (M^+^); Anal. Calcd for C_24_H_15_NO_5_: C, 72.54; H, 3.80; N, 3.52; found: C, 72.47; H, 3.79; N, 3.45.

#### 3.12.11. 6-Nitro-1-Phenyl-3-(phenylsulfonyl)-2-naphthol (**28**)

Yield: 0.43 g (89%) as a yellow solid, m.p. 155–156 °C; IR: 3498, 1344, 1150 cm^−1^; ^1^H NMR (400 MHz, CDCl_3_): δ 8.85 (d, *J* = 2.4 Hz, 1H), 8.70 (s, 1H), 8.56 (s, 1H), 8.16 (dd, *J* = 9.4, 2.3 Hz, 1H), 8.07–8.04 (complex, 2H), 7.65 (m, 1H), 7.59–7.53 (complex, 6H), 7.35–7.32 (complex, 2H); ^13^C NMR (101 MHz, CDCl_3_): δ 150.3, 144.5, 140.5, 139.0, 134.3, 133.1, 132.8, 130.6, 129.6, 129.12, 129.06, 128.8, 127.6, 127.1, 126.5, 126.2, 125.8, 122.5; MS (*m*/*z*): 405 (M^+^); Anal. Calcd for C_22_H_15_NO_5_S: C, 65.18; H, 3.73; N, 3.45; found: C, 65.09; H, 3.72; N, 3.39.

#### 3.12.12. 5-Nitro-2-(Phenylsulfonyl)benzaldehyde (**29**)

The dinucleophile 1,3-bis(phenylsulfonyl)propan-2-one (**17**) decomposed under the standard conditions (2 equiv. of K_2_CO_3_ in DMF at 65–70 °C). When Et_3_N (2 equiv.) was employed as the base in DMF at the same temperature, **29** was produced in a good yield after 30 min. Yield: 0.38 g (82%) as a white solid, m.p. 94–95 °C; IR: 1698, 1531, 1344, 1324, 1153 cm^−1^; ^1^H NMR (400 MHz, CDCl_3_): δ 10.90 (s, 1H), 8.80 (d, *J* = 2.4 Hz, 1H), 8.55 (dd, *J* = 8.6, 2.4 Hz, 1H), 8.36 (d, *J* = 8.6 Hz, 1H), 7.93 (d, *J* = 7.8 Hz, 2H), 7.70 (tt, *J* = 7.7, 1.3 Hz, 1H), 7.61 (t, *J* = 7.7 Hz, 2H); ^13^C NMR (101 MHz, CDCl_3_): δ 187.0, 150.8, 147.4, 140.0, 135.5, 134.7, 131.2, 130.1, 127.9, 127.7, 124.6; MS (*m*/*z*): 291 (M^+^); Anal. Calcd for C_13_H_9_NO_5_S: C, 53.61; H, 3.11; N, 4.81; found: C, 53.65; H, 3.13; N, 4.77. The product was independently prepared by treating **5a** with 2 equiv. of **17** in DMF using 2 equiv. of Et_3_N at 65–70 °C for 30 min. The spectral data were identical.

#### 3.12.13. Dimethyl 6-Cyano-2-hydroxynaphthalene-1,3-dicarboxylate (**30**)

Yield: 0.32 g (84%) as a white solid, m.p. 183–184 °C; IR: 3220–2975, 2228, 1732, 1671 cm^−1^; ^1^H NMR (400 MHz, CDCl_3_): δ 11.50 (s, 1H), 8.60 (s, 1H), 8.20 (d, *J* = 1.7 Hz, 1H), 7.99 (d, *J* = 8.9 Hz, 1H), 7.69 (dd, *J* = 8.9, 1.7 Hz, 1H), 4.08 (s, 3H), 4.07 (s, 3H); ^13^C NMR (101 MHz, CDCl_3_): δ 168.9, 167.1, 157.6, 135.72, 135.65, 135.6, 130.4, 125.42, 125.38, 118.5, 116.3, 115.6, 108.2, 53.3, 52.9; MS (*m*/*z*): 285 (M^+^); Anal. Calcd for C_15_H_11_NO_5_: C, 63.16; H, 3.89; N, 4.91; found: C, 63.13; H, 3.87; N, 4.85.

#### 3.12.14. Methyl 7-Cyano-3-hydroxy-4-phenyl-2-naphthoate (**31**)

Yield: 0.34 g (83%) as a white solid, m.p. 190–191 °C; IR: 3240–2900, 2227, 1653 cm^−1^; ^1^H NMR (400 MHz, CDCl_3_): δ 13.10 (s, 1H), 8.84 (d, *J* = 9.1 Hz, 1H), 8.14 (d, *J* = 1.9 Hz, 1H), 7.96 (s, 1H), 7.71 (dd, *J* = 9.1, 1.9 Hz, 1H), 7.62 (d, *J* = 7.4 Hz, 2H), 7.53–7.42 (complex, 3H), 4.17 (s, 3H); ^13^C NMR (101 MHz, CDCl_3_): δ 172.7, 164.6, 136.6, 136.1, 134.5, 134.0, 133.3, 129.5, 129.0, 128.4, 128.3, 127.4, 126.4, 119.0, 107.5, 105.2, 53.0; MS (*m*/*z*): 303 (M^+^); Anal. Calcd for C_19_H_13_NO_3_: C, 75.24; H, 4.32; N, 4.62; found: C, 75.23; H, 4.29; N, 4.57.

#### 3.12.15. Methyl 7-Cyano-4-(2-fluorophenyl)-3-hydroxy-2-naphthoate (**32**)

Yield: 0.40 g (92%) as an orange solid, m.p. 179–180 °C; IR: 3205–2895, 2234, 1724 cm^−1^; ^1^H NMR (400 MHz, CDCl_3_): δ 8.05 (td, *J* = 7.7, 1.8 Hz, 1H), 7.56 (dd, *J* = 8.5, 2.0 Hz, 1H), 7.42 (d, *J* = 2.0 Hz, 1H), 7.32 (s, 1H), 7.25–7.19 (complex, 1H), 7.15 (td, *J* = 7.6, 1.5 Hz, 1H), 7.10–7.04 (complex, 1H), 7.04 (d, *J* = 8.5 Hz, 1H), 6.88 (s, 1H), 3.92 (s, 3H); ^13^C NMR (101 MHz, CDCl_3_): δ 163.9, 160.1 (d, *J* = 250.1 Hz), 156.5, 144.8 (d, *J* = 2.5 Hz), 135.8, 131.9, 130.4, 129.6 (d, *J* = 2.5 Hz), 128.5 (d, *J* = 8.5 Hz), 125.6, 123.8 (d, *J* = 3.6 Hz), 122.6 (d, *J* = 11.7 Hz), 120.8, 118.0, 116.3, 115.2 (d, *J* = 22.5 Hz), 106.9, 100.7 (d, *J* = 7.8 Hz), 52.8; MS (*m*/*z*): 321 (M^+^); Anal. Calcd for C_19_H_12_FNO_3_: C, 71.03; H, 3.76; N, 4.36; found: C, 70.97; H, 3.78; N, 4.33.

#### 3.12.16. Methyl 4-(4-Chlorophenyl)-7-cyano-3-hydroxy-2-naphthoate (**33**)

Yield: 0.42 g (92%) as a white solid, m.p. 240–241 °C; IR: 3230–2875, 2227, 1718 cm^−1^; ^1^H NMR (400 MHz, CDCl_3_): δ 13.12 (s, 1H), 8.84 (d, *J* = 9.1 Hz, 1H), 8.14 (d, *J* = 1.9 Hz, 1H), 7.94 (s, 1H), 7.71 (dd, *J* = 9.1, 1.9 Hz, 1H), 7.57 (d, *J* = 8.4 Hz, 2H), 7.47 (d, *J* = 8.4 Hz, 2H), 4.17 (s, 3H); ^13^C NMR (101 MHz, CDCl_3_): δ 172.6, 164.3, 136.5, 134.52, 134.49, 134.45, 133.4, 132.8, 130.9, 129.2, 128.6, 127.4, 126.5, 118.9, 107.5, 105.4, 53.1; MS (*m*/*z*): 337, 339 (*ca*. 3:1, M^+^); Anal. Calcd for C_19_H_12_ClNO_3_: C, 67.57; H, 3.58; N, 4.15; found: C, 67.54; H, 3.56; N, 4.08.

#### 3.12.17. Methyl 7-Cyano-3-hydroxy-4-(4-methoxyphenyl)-2-naphthoate (**34**)

Yield: 0.42 g (94%) as a white solid, m.p. 198–199 °C; IR: 3184–2895, 2838, 1719 cm^−1^; ^1^H NMR (400 MHz, CDCl_3_): δ 13.12 (s, 1H), 8.82 (d, *J* = 9.1 Hz, 1H), 8.13 (d, *J* = 1.9 Hz, 1H), 7.93 (s, 1H), 7.68 (dd, *J* = 9.1, 1.9 Hz, 1H), 7.57 (d, *J* = 8.2 Hz, 2H), 7.03 (d, *J* = 8.2 Hz, 2H), 4.16 (s, 3H), 3.88 (s, 3H); ^13^C NMR (101 MHz, CDCl_3_): δ 172.8, 164.7, 159.7, 136.6, 134.4, 133.6, 133.1, 130.7, 128.8, 128.4, 127.5, 126.4, 119.1, 113.9, 107.3, 105.2, 55.4, 53.0; MS (*m*/*z*): 333 (M^+^); Anal. Calcd for C_20_H_15_NO_4_: C, 72.06; H, 4.54; N, 4.20; found: C, 72.02; H, 4.54; N, 4.14.

#### 3.12.18. Methyl 7-Cyano-3-hydroxy-4-(2,5-dimethylphenyl)-2-naphthoate (**35**)

Yield: 0.42 g (95%) as an orange solid, m.p. 164–165 °C; IR: 3210–2900, 2225, 1717 cm^−1^; ^1^H NMR (400 MHz, CDCl_3_): δ 7.68 (s, 1H), 7.52 (dd, *J* = 8.4, 2.0 Hz, 1H), 7.39 (d, *J* = 2.0 Hz, 1H), 7.27 (s, 1H), 7.10 (d, *J* = 7.6 Hz, 1H), 6.99 (d, *J* = 7.6 Hz, 1H), 6.92 (d, *J* = 8.4 Hz, 1H), 6.80 (s, 1H), 3.91 s, 3H), 2.36 (s, 3H), 2.32 (s, 3H); ^13^C NMR (101 MHz, CDCl_3_): δ 164.1, 156.9, 143.3, 135.8, 134.8, 133.7, 132.9, 131.7, 130.01, 129.99, 129.3, 128.0, 125. 7, 121.0, 118.2, 116.2, 107.4, 106.5, 52.6, 21.3, 20.0; MS (*m*/*z*): 331 (M^+^); Anal. Calcd for C_21_H_17_NO_3_: C, 71.99; H, 4.89; N, 4.00; found: C, 72.00; H, 4.88; N, 3.94.

#### 3.12.19. 5,7 Dibenzoyl-6-hydroxy-2-naphthonitrile (**36**)

Yield: 0.40 g (80%) as a yellow solid, m.p. 135–136 °C; IR: 3250–2940, 2226, 1662, 1631 cm^−1^; ^1^H NMR (400 MHz, CDCl_3_): δ 11.61 (s, 1H), 8.37 (s, 1H), 8.23 (d, *J* = 1.6 Hz, 1H), 7.92 (m, 2H), 7.80 (m, 2H), 7.73 (tt, *J* = 7.5, 1.3 Hz, 1H), 7.68–7.58 (complex, 5H), 7.49 (t, *J* = 7.3 Hz, 2H); ^13^C NMR (101 MHz, CDCl_3_): δ 201.0, 195.6, 156.9, 138.1, 137.2, 136.9, 136.2, 136.1, 134.2, 133.3, 130.6, 129.68, 129.65, 128.94, 128.88, 125.4, 125.3, 123.0, 121.8, 118.4, 108.2; MS (*m*/*z*): 377 (M^+^); Anal. Calcd for C_25_H_15_NO_3_: C, 79.56; H, 4.01; N, 3.71; found: C, 79.49; H, 4.00; N, 3.69.

#### 3.12.20. Ethyl (E)-3-(2-Fluoro-5-(trifluoromethyl)phenyl)prop-2-enoate (**37**)

Yield: 0.17 g (63%) as a colorless oil; IR: 1723, 1642, 1333 cm^−1^; ^1^H NMR (400 MHz, CDCl_3_): δ 7.81 (m, 1H), 7.80 (d, *J* = 16.2 Hz, 1H), 7.62 (m, 1H), 7.33 (t, *J* = 9.2 Hz, 1H), 6.60 (d, *J* = 16.2 Hz, 1H), 4.29 (q, *J* = 7.1 Hz, 2H), 1.35 (t, *J* = 7.1 Hz, 3H); ^13^C NMR (101 MHz, CDCl_3_): δ 166.2, 162.7 (d, *J* = 259.2 Hz), 135.5 (d, *J* = 2.8 Hz), 124.8 (dq, *J* = 9.6, 3.8 Hz), 127.3 (q, *J* = 33.4 Hz), 126.4 (dq, *J* = 7.8, 3.9 Hz), 123.5 (q, *J* = 271.9 Hz), 123.3 (d, *J* = 12.9 Hz), 122.8 (d, *J* = 6.4 Hz), 117.0 (d, *J* = 23.4 Hz), 60.9, 14.3; MS (*m*/*z*): 262 (M^+^); Anal. Calcd for C_12_H_10_F_4_O_2_: C, 54.97; H, 3.84; found: C, 54.99; H, 3.87.

#### 3.12.21. Methyl 3-Hydroxy-4-(4-methylphenyl)-7-trifluoromethyl-2-naphthoate (**38**)

Yield: 0.34 g (90%) as an orange solid, m.p. 248–249 °C; IR: 3518, 1722, 1333, 1210, 1172 cm^−1^; ^1^H NMR (400 MHz, CDCl_3_): δ 7.62 (d, *J* = 8.2 Hz, 2H), 7.51 (dd, *J* = 8.6, 2.3 Hz, 1H), 7.36 (d, *J* = 2.3 Hz, 1H), 7.27 (s, 1H), 7.18 (d, *J* = 8.2 Hz, 2H), 7.06 (d, *J* = 8.6 Hz, 1H), 6.59 (s, 1H), 3.89 (s, 3H), 2.36 (s, 3H); ^13^C NMR (101 MHz, CDCl_3_): δ 164.4, 156.1, 143.7, 136.7, 132.1, 130.4, 129.09, 129.08, 128.9 (q, *J* = 3.7 Hz), 125.3, 125.24 (q, *J* = 33.3 Hz), 125.16 (q, *J* = 3.8 Hz), 123.7 (q, *J* = 271.9 Hz), 120.1, 115.4, 108.6, 52.5, 21.3; MS (*m*/*z*): 360 (M^+^); Anal. Calcd for C_20_H_15_F_3_O_3_: C, 66.67; H, 4.20; found: C, 66.63; H, 4.17.

#### 3.12.22. Methyl 3-Hydroxy-4-(4-methoxylphenyl)-7-(trifluoromethyl)-2-naphthoate (**39**)

Yield: 0.37 g (88%) as an orange solid, m.p. 186–187 °C; IR: 3189–2877, 2838, 1714, 1318, 1208, 1112 cm^−1^; ^1^H NMR (400 MHz, CDCl_3_): δ 7.67 (d, *J* = 8.9 Hz, 2H), 7.49 (dd, *J* = 8.5, 2.2 Hz, 1H), 7.34 (d, *J* = 2.2 Hz, 1H), 7.23 (s, 1H), 7.04 (d, *J* = 8.5 Hz, 1H), 6.91 (d, *J* = 8.9 Hz, 2H), 6.57 (s, 1H), 3.89 (s, 3H), 3.84 (s, 3H); ^13^C NMR (101 MHz, CDCl_3_): δ 164.4, 158.5, 156.2, 143.0, 130.5, 129.8, 128.8 (q, *J* = 3.7 Hz), 127.8, 125.4, 125.2 (q, *J* = 33.1 Hz), 125.1 (m), 123.7 (q, *J* = 272.1 Hz), 120.3, 115.3, 113.8, 108.3, 55.3, 52.5; MS (*m*/*z*): 376 (M^+^); Anal. Calcd for C_20_H_15_F_3_O_4_: C, 63.83; H, 4.02; found: C, 63.82; H, 3.99.

#### 3.12.23. Ethyl (E)-3-(2-Fluoropyridin-3-yl)prop-2-enoate (**40**)

Yield: 0.23 g (72%) as a white solid, m.p. 44–45 °C; IR: 1723, 1645, 1327, 1243 cm^−1^; ^1^H NMR (400 MHz, CDCl_3_): δ 8.22 (dd, *J* = 4.9, 1.6 Hz, 1H), 7.95 (ddd, *J* = 9.4, 7.5, 1.0 Hz, 1H), 7.73 (d, *J* = 16.2 Hz, 1H), 7.26 (m, 1H), 6.60 (d, *J* = 16.2 Hz, 1H), 4.28 (q, *J* = 7.1 Hz, 2H), 1.35 (t, *J* = 7.1 Hz, 3H); ^13^C NMR (101 MHz, CDCl_3_): δ 166.2, 161.3 (d, *J* = 245.0 Hz), 148.6 (d, *J* = 15.4 Hz), 139.4 (d, *J* = 3.7 Hz), 135.6 (d, *J* = 3.0 Hz), 123.1 (d, *J* = 6.3 Hz), 121.9 (d, *J* = 4.5 Hz), 117.7 (d, *J* = 26.7 Hz), 60.9, 14.3; MS (*m*/*z*): 195 (M^+^); Anal. Calcd for C_10_H_10_FNO_2_: C, 61.53; H, 5.16; N, 7.18; found: C, 61.48; H, 5.13; N, 7.11.

#### 3.12.24. Methyl 7-Hydroxy-8-phenylquinoline-6-carboxylate (**41**)

Yield: 0.39 g (88%) as an orange oil; IR: 3140–2975, 1725 cm^−1^; ^1^H NMR (400 MHz, CDCl_3_): δ 8.20 (dd, *J* = 5.0, 2.0 Hz, 1H), 7.86 (d, *J* = 7.4 Hz, 2H), 7.44 (dd, *J* = 7.4, 1.9 Hz, 1H), 7.38 (t, *J* = 7.7 Hz, 2H), 7.28 (s, 1H), 7.22 (tt, *J* = 7.4, 1.3 Hz, 1H), 6.98 (dd, *J* = 7.4, 5.0 Hz, 1H), 6.64 (s, 1H), 3.90 (s, 3H); ^13^C NMR (101 MHz, CDCl_3_): δ 164.3, 159.8, 150.0, 144.8, 136.3, 134.8, 130.8, 129.5, 128.5, 127.0, 125.4, 119.9, 115.3, 109.4, 52.5; MS (*m*/*z*): 279 (M^+^); Anal. Calcd for C_17_H_13_NO_3_: C, 73.11; H, 4.69; N, 5.02; found: C, 73.07; H, 4.66; N, 5.01.

#### 3.12.25. Methyl 8-(4-Chlorophenyl)-7-hydroxyquinoline-6-carboxylate (**42**)

Yield: 0.43 g (86%) as an orange solid, m.p. 134–135 °C; IR: 3265–2885, 1721 cm^−1^; ^1^H NMR (400 MHz, CDCl_3_): δ 8.21 (dd, *J* = 5.0, 1.9 Hz, 1H), 7.79 (d, *J* = 8.6 Hz, 2H), 7.46 (dd, *J* = 7.3, 1.9, 1H), 7.33 (d, *J* = 8.6 Hz, 2H), 7.31 (s, 1H), 7.00 (d, *J* = 8.6 Hz, 1H), 6.62 (s, 1H), 3.90 (s, 3H); ^13^C NMR (101 MHz, CDCl_3_): δ 164.2, 159.6, 150.1, 145.2, 136.4, 133.3, 132.4, 131.2, 130.6, 128.6, 125.0, 120.1, 115.2, 108.1, 52.6; MS (*m*/*z*): 313, 315 (*ca*. 3:1, M^+^); Anal. Calcd for C_17_H_12_ClNO_3_: C, 65.08; H, 3.86; N, 4.46; found: C, 65.11; H, 3.88; N, 4.38.

#### 3.12.26. Methyl 2-Hydroxy-8-(4-methylphenyl)quinoline-1-carboxylate (**43**)

Yield: 0.41 g (88%) as an orange solid, m.p. 121–122 °C; IR: 3330–2897, 1724 cm^−1^; ^1^H NMR (400 MHz, CDCl_3_): δ 8.18 (dd, *J* = 5.0, 1.9 Hz, 1H), 7.76 (d, *J* = 8.2 Hz, 2H), 7.42 (dd, *J* = 7.3, 1.9 Hz, 1H), 7.23 (s, 1H), 7.19 (d, *J* = 8.2 Hz, 2H), 6.96 (dd, *J* = 7.3, 5.0 Hz, 1H), 6.59 (s, 1H), 3.90 (s, 3H), 2.35 (s, 3H); ^13^C NMR (101 MHz, CDCl_3_): δ 164.4, 159.9, 149.9, 144.3, 136.9, 136.1, 131.9, 130.3, 129.4, 129.2, 125.6, 119.8, 115.4, 109.4, 52.5, 21.4; MS (*m*/*z*): 293 (M^+^); Anal. Calcd for C_18_H_15_NO_3_: C, 73.71; H, 5.15; N, 4.78; found: C, 73.67; H, 5.13; N, 4.75.

#### 3.12.27. Methyl 7-Hydroxy-8-(4-methoxyphenyl)quinoline-6-carboxylate (**44**)

Yield: 0.42 g (84%) as an orange solid, m.p. 103–104 °C; IR: 3196–2893, 2841, 1719 cm^−1^; ^1^H NMR (400 MHz, CDCl_3_): δ 8.14 (dd, *J* = 5.0, 1.9 Hz, 1H), 7.81 (d, *J* = 8.8 Hz, 2H), 7.37 (dd, *J* = 7.3, 1.9 Hz, 1H), 7.17 (s, 1H), 6.92 (coincident d, *J* = 8.8 Hz, 2H and m, 1H), 6.57 (s, 1H), 3.87 (s, 3H), 3.82 (s, 3H); ^13^C NMR (101 MHz, CDCl_3_): δ 164.4, 159.9, 158.5, 149.7, 143.6, 136.1, 130.9, 129.8, 127.7, 123.5, 119.8, 115.5, 113.9, 109.0, 55.2, 52.5; MS (*m*/*z*): 309 (M^+^); Anal. Calcd for C_18_H_15_NO_4_: C, 69.89; H, 4.89; N, 4.53; found: C, 69.84; H, 4.88; N, 4.51.

## 4. Conclusions

In conclusion, we developed a new strategy for the synthesis of trisubstituted 2-naphthol and disubstituted 7-hydroxyquinoline derivatives. The sequence involves a reaction between 1 equiv. of an S_N_Ar-activated 2-fluorobenzaldehyde (dielectrophile) and 2 equiv. of a 1,3-disubstituted acetone (dinucleophile). The use of 2-fluoronicotinaldehyde as the dielectrophile yields 7-hydroxyquinoline products. The transformation occurs in a single flask and requires only 2 equiv. of K_2_CO_3_ in anhydrous DMF at 65–70 °C. Thus, extensive process optimization is not required. Following mild acid workup and column chromatography, the yields are high. The reaction is regioselective with the most stable anionic center attacking the aldehyde first, followed by S_N_Ar ring closure from the less stable anion. The current approach avoids the use of strong acids or bases under protic conditions which can attack other functionality. Additionally, expensive metal catalysts which can be difficult to remove from the final product are not required. We are continuing our work to elucidate additional [3+3] approaches to privileged ring systems important in drug synthesis.

## Data Availability

No new data were created or analyzed in this study. Data sharing is not applicable to this article.

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
