# Peer review of "A [3+3] Aldol-SNAr-Dehydration Approach to 2-Naphthol and 7-Hydroxyquinoline Derivatives"

_molecules, 2024, doi:10.3390/molecules29143406_

Round 1

Reviewer 1 Report

Comments and Suggestions for Authors

The manuscript Molecules-3108500 titled “A [3+3] Aldol-SNAr-Dehydration Approach to 2-Naphthol and 2 7-Hydroxyquinoline Derivatives” submitted by R.A. Bunce et al. presents a novel and efficient approach for the synthesis of 2-naphthol and 7-hydroxyquinoline derivatives via a [3+3]aldol-SNAr-dehydration annulation sequence.

The introduction is well structured and documented. However, the results and discussions, although they are good results, are presented in a disorderly manner and lack an in-depth analysis of the interesting process presented. The conclusions are consistent with the results shown.

In my opinion, this manuscript may be reconsidered after major revisions, which should include the following:

1- In the results and discussions (or supplementary material) it is not possible to observe the origin of the reaction conditions established in this study. Therefore, it is essential to carry out a thorough process optimization analysis, which includes the influence of bases (carbonates, phosphates, amines, etc.), solvents, temperatures, reaction stoichiometry, reaction molarity, effect of solvent humidity. , among others. These data must be included in a table in the results and discussions section and present the respective analyzes that allow establishing the optimal reaction conditions in a model substrate.

2- The authors present tables 1 to 4, but do not give greater influence in the discussions of the results presented here. The discussions corresponding to each table must be organized and the main disadvantages in each case analyzed. Please do not incorporate tables consecutively without a prior discussion text. It is recommended to include "entries" in the tables (e.g. Table 1, entry 1) to make the sections easier to read.

3- In Materials and methods, the authors present subsections 3.2 to 3.11 with information on the characterization of starting substances. However, the synthesis of these products is not present. Please include them in each case (group the cases as general methods if possible).

Author Response

Reviewer 1

The manuscript Molecules-3108500 titled “A [3+3] Aldol-SNAr-Dehydration Approach to 2-Naphthol and 2 7-Hydroxyquinoline Derivatives” submitted by R.A. Bunce et al. presents a novel and efficient approach for the synthesis of 2-naphthol and 7-hydroxyquinoline derivatives via a [3+3]aldol-SNAr-dehydration annulation sequence.

The introduction is well structured and documented. However, the results and discussions, although they are good results, are presented in a disorderly manner and lack an in-depth analysis of the interesting process presented. The conclusions are consistent with the results shown.

In my opinion, this manuscript may be reconsidered after major revisions, which should include the following:

Comment 1- In the results and discussions (or supplementary material) it is not possible to observe the origin of the reaction conditions established in this study. Therefore, it is essential to carry out a thorough process optimization analysis, which includes the influence of bases (carbonates, phosphates, amines, etc.), solvents, temperatures, reaction stoichiometry, reaction molarity, effect of solvent humidity, among others. These data must be included in a table in the results and discussions section and present the respective analyzes that allow establishing the optimal reaction conditions in a model substrate.

Response 1- Several of our recent papers have all found that the temperature, equivalents of base and run time are similar for the SNAr-terminated reactions we have studied.  The only difference in this paper is that the disubstituted acetone (dinucleophile) needed to be used in 2-fold excess.  It is not exactly clear why this is, but adjustments to other reactants/reagents did not improve the yields and in fact lowered them. I have added a short comment on this in the manuscript.

Comment 2- The authors present tables 1 to 4, but do not give greater influence in the discussions of the results presented here. The discussions corresponding to each table must be organized and the main disadvantages in each case analyzed. Please do not incorporate tables consecutively without a prior discussion text. It is recommended to include "entries" in the tables (e.g. Table 1, entry 1) to make the sections easier to read.

Response 2-I have labeled the entries in the Tables

Comment 3- In Materials and methods, the authors present subsections 3.2 to 3.11 with information on the characterization of starting substances. However, the synthesis of these products is not present. Please include them in each case (group the cases as general methods if possible).

Response 3- The exact same procedure and conditions were used for each of the substrates.  Only Table 1, entry 12 and Table 3 entry 1 and Table 4 entry 1 gave unexpected results.  References for the procedures to prepare the starting materials are given in the text and (for some) at the beginning of the General Methods Section. 

Reviewer 2 Report

Comments and Suggestions for Authors

Manuscript titled “A [3+3] Aldol-SNAr-Dehydration Approach to 2-Naphthol and 7-Hydroxyquinoline Derivatives” by Bunne et al. describes an easy and straigtforward synthesis of 2-Naphthol and 7-Hydroxyquinoline derivatives using 1,3-disubstituted acetone analogues and 2-fluorobenzaldehyde derivatives without any additives or metal catalysts.

1. Literature reports suggest that some of these 2-naphthol-based drug candidates have been expressed significant activity against cancer as well as Gram-positive and Gram-negative bacteria.
2. Current protocol avoids reaction optimization and problems associated with postreaction metal contamination.
3. References are appropriate.
4. It would be beneficial to add details to Figure 1, explaining what numbers 1, 2, 3, and 4 represent. I recommend including the names and activities of each.
Although these details are mentioned in the text, displaying them in the figure will enhance visibility and interest to readers.

The manuscript is well-written and suited for “Molecules”. It discusses in detail about substrate 1,3-bis(phenylsulfonyl)propan-2-one which failed to provide the expected product. With that said, I recommend acceptance of the manuscript after minor revisions.

Author Response

Reviewer 2

Manuscript titled “A [3+3] Aldol-SNAr-Dehydration Approach to 2-Naphthol and 7-Hydroxyquinoline Derivatives” by Bunce et al. describes an easy and straightforward synthesis of 2-Naphthol and 7-Hydroxyquinoline derivatives using 1,3-disubstituted acetone analogues and 2-fluorobenzaldehyde derivatives without any additives or metal catalysts.

  1. Literature reports suggest that some of these 2-naphthol-based drug candidates have been expressed significant activity against cancer as well as Gram-positive and Gram-negative bacteria.  Response 1- No changes needed.

  2. Current protocol avoids reaction optimization and problems associated with postreaction metal contamination.  Response 2- No changes needed.

  3. References are appropriate.  Response 3- No changes needed

  4. It would be beneficial to add details to Figure 1, explaining what numbers 1, 2, 3, and 4 represent. I recommend including the names and activities of each. 

Although these details are mentioned in the text, displaying them in the figure will enhance visibility and interest to readers.  Response 4-  This has been done.

Done

The manuscript is well-written and suited for “Molecules”. It discusses in detail about substrate 1,3-bis(phenylsulfonyl)propan-2-one which failed to provide the expected product. With that said, I recommend acceptance of the manuscript after minor revisions.

Reviewer 3 Report

Comments and Suggestions for Authors

I think this article could be published after major revisions. It is mostly well done, and the conclusions are supported by experimental data, however, there are some issues that in my opinion the authors should address.

1)      Firstly, while it is true that benzaldehydes 5a-d are commercially available their cost is rather prohibitive, so is it possible to use simple o-nitrobenzaldehyde instead of C5-EWG-substituted o-fluorobenzaldehyde in this transformation? It would be nice if the authors demonstrated this with at least one or two examples to enhance the practicality of the given protocol.

2)      Although the authors claim that 2 equiv of 1,3-disubstituted acetones 6-17 give the best yields of naphthols 18-44, it would still be useful to see some optimization data. The synthesis of compounds 6-17 does not appear to be straightforward, and maybe using them in slight excess or equimolar amounts would be a good trade-off for the presumably lower yields of the target naphthols.

3)      In the experimental part for the synthesis of acetone derivatives 6-17, please, provide the scale/absolute quantities (in g/mg) of the latter.

4)      Please, provide the 19F NMR data for all fluorocontaining compounds (as 9, 13, 21 etc)

5)      In SI, p 3, there is a type error 3-oxo-4-pjhenylbutanoate

6)      In SI, p 6, should be methyl 4-(43-methylphenyl)-3-oxobutanoate

7)      In SI, the intensity of peaks both proton and carbon NMR in almost all spectra is too low, it is difficult to see anything, should be increased so they occupy about 30% of page/picture

8)      In SI, some 13C spectra (22, 33 etc) have extra peaks labeled as acetone, if so, then the corresponding 1H spectra should same label has to be in as well. Also, it is not clear where it came from, since no acetone was used according to the experimental procedure

Author Response

Reviewer 3

I think this article could be published after major revisions. It is mostly well done, and the conclusions are supported by experimental data, however, there are some issues that in my opinion the authors should address.

Commnet 1- Firstly, while it is true that benzaldehydes 5a-d are commercially available their cost is rather prohibitive, so is it possible to use simple o-nitrobenzaldehyde instead of C5-EWG-substituted o-fluorobenzaldehyde in this transformation? It would be nice if the authors demonstrated this with at least one or two examples to enhance the practicality of the given protocol

Response 1- This would be a different reaction and would not work since the ring is not activated toward the final SNAr ring closure.  Our substrates are not overly expensive at the exploratory scales we are using. 

Comment 2- Although the authors claim that 2 equiv of 1,3-disubstituted acetones 6-17 give the best yields of naphthols 18-44, it would still be useful to see some optimization data. The synthesis of compounds 6-17 does not appear to be straightforward, and maybe using them in slight excess or equimolar amounts would be a good trade-off for the presumably lower yields of the target naphthols.

Response 2- My students have done many of these types of reactions. Without much experimentation, they found that the reaction worked best when 2 equiv of the disubstituted acetones (dinucleophiles) were used.  Normally, I do not like to waste reagents that take time and effort to prepare, but this is what we observed.  The use of 1:1 quantities of the two reagents and attempts to increase the amount of the dielectrophile all gave lower yields.  A short comment on this has been added to the text.

Comment 3- In the experimental part for the synthesis of acetone derivatives 6-17, please, provide the scale/absolute quantities (in g/mg) of the latter.

 Response 3- This has been done.

Comment 4-  Please, provide the 19F NMR data for all fluorocontaining compounds (as 9, 13, 21 etc).

Response 4- No additional compound samples are available for these spectra.

Comment 5- In SI, p 3, there is a type error 3-oxo-4-pjhenylbutanoate

Response 5-  Corrected, thank you.

Comment 6- In SI, p 6, should be methyl 4-(4-methylphenyl)-3-oxobutanoate

Response 6- Corrected, thank you.

Round 2

Reviewer 1 Report

Comments and Suggestions for Authors

The authors have made sufficient changes as requested to support publication in its current form.

Reviewer 3 Report

Comments and Suggestions for Authors

I think the article can be published as it is. I’m still thinking that o-nitrobenzaldehydes would work just fine on their own. There are plenty of examples of cyclization reactions caused by ipso-substitution at the nitro group, for instance, 10.1021/acs.joc.3c00134. Also, the statement that the ratio of reagents other than 2 to 1 gives lower yields, to be honest, is not very informative. If not a full-scale optimization table, why not just list the obtained yields in parentheses, like, for example, “…other combinations (1:1, 1.5:1, 2:1, and 1:1.5) of regents gave lower yields, namely, 55%, 67%, 93%, and 53%, respectively…” Let the readers decide for themselves which of the low values is significant to them or not. Otherwise, as i said earlier, the work is generally well done and presented. Therefore, it is fully suitable for publication in the journal.